# Prevalence and factors associated with delayed antiretroviral therapy initiation among adults with HIV in Alebtong district, Northern Uganda: A facility-based study

Anthony Mark Ochen[1]☯*, David Lubogo[2‡], Michael Ediau[1,2,3,4‡], Victoria Nankabirwa[5☯]

1 Department of Health Policy, Planning, and Management, School of Public Health, College of Health Sciences, Makerere University, Kampala, Uganda, 2 Department of Community Health and Behavioural Sciences, School of Public Health, College of Health Sciences, Makerere University, Kampala, Uganda, 3 Division of Epidemiology and Biostatistics, School of Public Health, San Diego State University, San Diego, CA, United States of America, 4 Division of Global Public Health, Department of Family Medicine and Public Health, School of Medicine, University of California San Diego, La Jolla, CA, United States of America, 5 Department of Epidemiology and Biostatistics, School of Public Health, College of Health Sciences, Makerere University, Kampala, Uganda

☯ These authors contributed equally to this work.
‡ DL and ME also contributed equally to this work.
* markochen@rocketmail.com

**Data Availability Statement:** All data are in the manuscript and/or supporting information files.

## Abstract

Globally, an estimated 36.7 million people were living with HIV (PLWH) and of these, 2.1 million were newly infected and 1.1 million died of AIDS in 2015. By 2016, only 67% of adults eligible for ART were enrolled in ART in Uganda. Delayed ART initiation has been shown to contribute to the continued transmission of HIV as well as to higher morbidity and mortality among persons living with HIV. Our study examined the prevalence and factors associated with delayed ART initiation among adults with HIV in Alebtong district, Northern Uganda. A cross-sectional study involving 432 adults living with HIV was conducted between March and June 2018 in Alebtong district. Quantitative data were collected using interviewer-administered questionnaires and desk reviews using a data extraction tool. A binary logistic regression using a hierarchical modelling technique was used at the multivariable level to determine associations at a 95% confidence interval and p<0.05 using SPSS Statistics software version 23.0. Overall, 432 participants were enrolled in the study, of whom 18.1% (78/432) had delayed ART initiation. After final adjustment, our key findings showed a significantly lower odds of delayed ART initiation among older respondents (aOR = 0.35, 95% CI: 0.16–0.76); adherence to HIV clinic appointments, (aOR = 0.06, 95% CI: 0.02–0.15); and linkage to the HIV clinic the same day HIV test was conducted (aOR = 0.21, 95% CI: 0.08–0.55). However, a significantly higher odds of delayed ART initiation was observed among those whose cultures do not support the use of ART (aOR = 10.62, 95% CI: 3.04–32.08). Reducing delayed ART initiation in the district requires strengthening the involvement of adolescents and young people in the HIVAIDS programming, scaling up the implementation of the same-day ART initiation policy, and addressing negative cultural beliefs affecting early ART initiation in the district.

**Funding:** This work was supported by the German DAAD Scholarship Programme (through the In-Country/In-Region Scholarships Programme Uganda, 2017 to AMO). The funders had no role in the study design, data collection, and analysis, decision to publish, or preparation of the manuscript.

**Competing interests:** The authors have declared that no competing interests exist.

## Introduction

Globally, an estimated 36.7 million people were living with HIV and of these, 2.1 million were newly infected and 1.1 million died of Acquired Immune-deficiency Syndrome (AIDS) in 2015 [1]. An estimated 24.7 million people living with HIV are in sub-Saharan Africa (SSA) representing nearly 71% of the global total. Uganda and Angola are the two countries where the number of new HIV infections increased by 21% between 2005 and 2013 [1].

Although it is estimated that the implementation of ART programmes has added 15 million life years globally [2], about 1.7 million people died of HIV-related illnesses in 2011, and more than 90% of them were living in low and middle-income countries [2]. However, ART was responsible for a 26% decline in AIDS-related deaths since 2010. [2]. While the number of people receiving ART has been quickly increasing in recent years, from 9.7 million in 2012 to 15.8 million in 2015 [3], the global ART coverage among adults was only 38% at the end of 2013 [4]. Even though ART is free in many low and middle-income countries including Uganda, studies from sub-Saharan Africa have shown that 25% of patients in need of ART do not start on time [2]. Much as the HIV care services have been increasingly scaled up, most persons living with HIV in sub-Saharan Africa start treatment after developing advanced infection [5]. The problem of delayed ART initiation in sub-Saharan Africa has improved only slightly since the start of the HIV scale-up in the region [6]. In 2016, the WHO issued guidelines recommending that countries adopt a policy of initiating ART immediately after confirmed HIV testing [7]. This guideline recommends that ART should be initiated within 7 days following a confirmed HIV testing and on the same day for those patients who are ready to start [8].

In Alebtong district, information on delayed ART initiation among adults is unknown. This problem can contribute to poor planning and implementation of ART services leading to a missed opportunity for prevention of HIV transmission and mortality related to HIV/AIDs. While the Uganda Ministry of Health (MoH) has improved the supply of antiretroviral drugs to the district, developed new ART policy guidelines, and scaled up ART services, there is still a gap in the unknown prevalence of delayed ART initiation among adults. Our study will provide information that can be used to address challenges leading to delayed ART initiation. Provide information for health managers, relevant stakeholders, and MoH to effectively plan for the implementation of ART services that will contribute to increased access to ART and early ART initiation. Furthermore, the health of the PLWH would improve and they live a healthy life. They would also become productive and contribute to the well-being of their families and societies. The information can also lead to programmatic and policy adjustments related to ART management that would benefit the PLWH. Therefore, our study examined the prevalence and factors associated with delayed ART initiation among adults living with HIV in Alebtong district, Northern Uganda.

## Methods and materials

### Ethics statement

The ethical approval to conduct the study was obtained from the Higher Degrees, Research and Ethics Committee (HDREC), Makerere University School of Public Health, Kampala (Uganda). Administrative authorization for the study was approved by Alebtong district health officer, in charge of health facilities and ART clinics, and written informed consent was obtained from each participant before the commencement of interviews. Confidentiality was maintained by selecting a private place for interviews, using secret codes instead of names of participants, and keeping data collected under lock-and-key. The non-ART initiators and loss to follow-up patients in the community were advised to seek medical care at the nearest health facility offering ART services.

## Study context

Our study was undertaken in Alebtong district in Northern Uganda. The district was identified based on the high HIV prevalence among the general population, 6.8% as compared to the national prevalence rate of 6.2% [9], coupled with reported poor documentation of ART initiation. Administratively, the district is sub-divided into two counties, eight sub-counties, 45 parishes, and 605 villages with a total estimated population of 227,530 people and 50,542 households [10]. The district has 18 health facilities of which only ten offers ART services. At least seven health facilities offering ART services were included in this study, five public and two private-not-for-profit (PNFP) health facilities.

In the early 1990s, Uganda was one of the worst-hit countries by the HIV/AIDS pandemic with a prevalence of 18%, however, a concerted effort through a multi-sectoral approach successfully reduced the HIV prevalence to 6.4% by 2005 [1]. The adult HIV prevalence stabilized at 6–7% between 2005 and 2010. In 2011, the country witnessed a resurgence in HIV prevalence rising to 7.3% among people aged 15–49 years [12]. Uganda is still classified as a high burden country with a high proportion of persons living with HIV [13]. The national HIV prevalence is estimated at 6.2% whereas the mid-Northern region and Alebtong district have an HIV prevalence of 7.2% and 6.8% respectively [14].

The scaling up of ART in Uganda gained momentum with three major global health initiatives: the Multi-Country HIV/AIDS Programme (MAP) in 2002; the United States President's Emergency Plan for AIDS Relief (PEPFAR) and the Global Fund to fight HIV/AIDS, Tuberculosis and Malaria (GFATM) in 2004 [15]. Free ARVs have been provided in public hospitals and health facilities since 2003 when the first national ART strategy and treatment guidelines were developed [16]. By the end of 2009, approximately 200,400 people were receiving ART and the coverage of those in need based on the WHO 2010 thresholds had reached 39% [17]. In 2014, around 1,660 health facilities were offering ART in Uganda, and nearly 751,000 people living with HIV were enrolled in treatment [13]. However, by the year 2016, only 67% of eligible adults were enrolled in ART [18].

In Uganda, preparing people to start ART is an important step to achieving ART success. Healthcare providers ought to initiate detailed discussions about the willingness and readiness of patients to initiate ART. The ART initiation is done at the earliest opportunity for all people with confirmed HIV regardless of clinical stage or $CD_4$ cell count. However, the process of starting ART starts with assessing all clients for opportunistic infections (OIs) especially tuberculosis (TB) and cryptococcal meningitis. For patients with TB or cryptococcal meningitis, ART is deferred and initiated after starting treatment for OIs but can also be started concurrently. For patients without TB or cryptococcal meningitis, ART is offered the same day through an opt-out approach. If the patient is ready, ART is initiated on the same day but if not ready or opt-out of the same-day initiation, a timely ART preparation plan is agreed upon to initiate ART within one month for those who are adults [16].

## Study design and sampling procedure

This was a facility-based cross-sectional study design that used a quantitative method of data collection and analysis. The sample size was determined using the Kish-Leslie formula [19], $n = \left( \frac{Z^2 pq}{d^2} \right)$ where n is the sample size, Zα is the standard normal deviate, set at 1.96 (for 95% confidence interval), p is the estimated target population with delayed ART initiation = 49% [14] and d is the desired degree of accuracy (taken as 0.05), giving a sample size of 348. But after adjusting for the 13.9% non-response rate [20] and invalid responses, the final sample size of 446 was calculated.

The study included adults who were; confirmed as living with HIV, 18 years and above, receiving ART services at a public or private-not-for profit health facilities accredited by the Uganda ministry of health to provide ART services, and who had consented to participate in the study. On the other hand, the study excluded adults confirmed as living with HIV who had transferred out to other districts, those who had refused to participate in the study, and those who had died. The seven health facilities offering ART services were randomly selected and probability proportional-to-size sampling was used to identify the required samples from each health facility. The respondents were identified using a systematic sampling method where the sampling frame was determined for each health facility (S1 Fig).

A total of 446 eligible participants were assigned for the survey, however, the final survey was administered to 432 participants who were confirmed with HIV in the previous 12 months, from April 2017 to March 2018. At least five patients refused to participate in the study, three had died and six had transferred to other districts. The sampling interval of two was determined by dividing the sampling frame of 831 patients who had initiated ART with the calculated sample size of 446. The first participant was randomly identified among the first two persons with HIV on the list and subsequent participants were identified by adding every second patient on the list. Those who declined to participate in the survey were excluded without replacements since non-response was already adjusted for in the calculation of sample size.

## Data collection tools and procedures

We identified seven Research Assistants (RAs) who were health workers with a qualification of bachelor's degrees and diplomas. They were also fluent in the local language of the study participants and had prior experience in HIV/AIDS-related surveys. The Research Assistants (RAs) were trained for two days on ethical procedures required during data collection and how to appropriately fill the data collection tools. The data collection tools were pre-tested for one day in Ogur HC IV and Bar HC III in Lira district. These health facilities were chosen because they had similar characteristics to those included in our survey. On the following day, all the survey team members converged and validated the tools.

Quantitative data was collected using a structured questionnaire and data extraction tool. The questions used in the survey were adapted from various sources including the Uganda Demographic and Health Survey (UDHS) questionnaire and other published research questionnaires related to this study. The tools were revised by the research supervisors, other academic staff, and peers. The structured questionnaire comprised of background information as well as individual, community, and health service factors (S1 Text). It was designed in English and later translated into the local language of the respondents (S2 Text) and then back-translated into English to facilitate easy data entry. The data extraction tool contained information on the date of HIV testing, ART initiation, linkage to HIV treatment and care, and the unit/department where clients were referred from (S1 Table).

Data collection was conducted between the periods of April 2017 to March 2018. The RAs administered the questionnaires using face-to-face interviews. On each clinic day (Monday to Friday), the RAs proceeded to selected health facilities and obtained the list of people living with HIV from the ART registers. The names of participants were arranged in Alphabetical order and systematically sampled to obtain the required samples. The eligible respondents present on each clinic day were sampled and informed consent was obtained from them before the commencement of interviews. Confidentiality was observed by using codes to identify participants and by selecting a private place where no other person could listen to the interview process. Data was collected using paper-based questionnaires where the RAs administered face-to-face interviews to each participant until all the required samples were completed. The

persons living with HIV in the community who were lost to follow-up were traced using their names, locations, and other details and were interviewed. The Principal Investigator (PI) conducted a document review using a data extraction tool after data collection was completed by the RAs. This aided in the tracing of codes for interviewed participants in the ART registries to record the date of confirmed HIV testing and ART initiation. The overall data collection took a period of three months, from March to May 2018.

## Variable selection

The selection of variables was based on a hierarchical conceptual framework used to perform multivariable analysis. Factors of interest were organised into four levels according to how they were thought to influence ART initiation among adults, adapted from Dahab et al, 2010 [21]. These factors included demographic factors; age, sex, education, occupation, place of residence, current marital status & religion; individual factors included: time taken to reach the health facility, frustration with HIV results, disclosure of HIV status, adherence to clinic appointments, alcohol consumption & satisfaction with ART services; community factors included: distance to nearest health facility offering ART services, socio-economic status of households, domestic violence, experience of stigma, cultural support; and health services factors included: HIV testing from the same facility where ART was initiated, received HIV result same day tested and attitude of health workers. Data extraction tool was used to collect data on; the date of HIV testing, ART initiation status, linkage to care same day tested for HIV & unit/department where patients were referred from.

## Outcome variable

Delayed ART initiation was the main outcome variable and was defined as adults (18 years and above) who were confirmed with HIV but initiated ART after 30 days of confirmed HIV testing irrespective of $CD_4$ counts and clinical staging [22]. It was categorized as a binary outcome where; delayed ART initiation was assigned a value of 1 and early ART initiation was assigned a value of 0.

## Data analysis

Data were double entered into SPSS Statistics version 23.0 software, cleaned, transformed, and analysed. Univariable analysis was performed to generate frequencies and proportions. Pearson's Chi-square ($x^2$) test was used to determine associations between independent and outcome variables at the bivariable level. This helped to identify variables included in the binary logistic regression analysis at the multivariable level. Independent variables that were found to be associated with the outcome variable were selected using a cut-off value of $p \leq 0.25$ and included in the multivariable analysis.

We used a hierarchal modelling technique to enter variables into the multivariable regression models. A total of four models were used in the multivariable analysis. Model 1 included demographic factors, model 2 individual factors, model 3 community factors, and model 4 health services factors. Model 1 was achieved by adding seven demographic variables; model 2 by adding six individual variables; model 3 by adding five community variables; and model 4 achieved by adding five health services variables. All the four models were simultaneously generated to produce a unified output, however, the fourth and last model was the one considered for explanations and discussions. This is because the last model had a rigorous adjustment using variables from the first, second, and third models so, it's considered the best model. Adjusted Odds Ratios (aORs) using a p-value of less than 0.05 and a 95% confidence interval were used to determine the significant association between independent and the outcome

variables. The multicollinearity diagnostic analysis was performed to eliminate variables in the models having a variance inflation factor (VIF) of more than six values. Hosmer and Lemeshow Chi-square statistics were used to determine the goodness of fit of data in the models.

## Results

### Participant's characteristics

Analysis of respondent's characteristics (N = 432) shows a higher proportion of respondents among females (60.9%), rural dwellers (93.8%), and younger respondents less than or equal to 35 years (67.8%). The majority of households (76.6%) had low socioeconomic status. More than one-half (56.9%) lived within five kilometres radius of the nearest health facility offering ART. Most respondents (64.6%) received ARV drugs the same day tested for HIV and 69.0% disclosed their HIV status after confirming their test results (Table 1).

### Prevalence of delayed ART initiation

The overall prevalence of delayed ART initiation among adults diagnosed with HIV in Alebtong district was 18.1% (Table 2). Of all the 432 respondents, a higher prevalence of delayed ART initiation (25.0%) was observed in Aloi Mission HC III (PNFP facility) and the lowest prevalence (10.5%) in Amugu HC III (public facility).

### Bivariable regression analysis of factors for delayed ART initiation

The unadjusted analysis of demographic, individual, community and health service factors with delayed ART initiation is shown in Table 3. The odds of delayed ART initiation were about twice among respondents who lived in urban areas as compared to rural dwellers (OR = 2.44; 95% CI: 1.05–5.65). Similarly, the odds of delayed ART initiation were approximately twice among women who were frustrated with HIV test results than the odds of those who were not (OR = 1.89; 95% CI: 1.13–3.18). Furthermore, the odds of delayed ART initiation were about eight times among respondents who said their culture supported ART as compared to those who said the contrary (OR = 8.18; 95% CI: 3.18–20.19). Additionally, the odds of delayed ART initiation among respondents who were linked to HIV care from community outreaches were approximately seven times more than the odds of those who were not (OR = 7.10; 95% CI: 3.40–14.84). However, lower odds of delayed ART initiation were observed among; respondents who disclosed their HIV status than those who did not (OR = 0.45; 95% CI: 027–0.74); respondents who adhered to health facility appointments as compared to those who did not (OR = 0.06; 95% CI: 0.03–0.13); respondents who never experienced domestic violence than those who sometimes experienced domestic violence (OR = 0.49; 95% CI: 0.25–0.97); respondents who were linked to HIV care the same day tested than those who were not (OR = 0.19; 95% CI: 0.11–0.33); respondents who felt health workers had a very good attitude towards persons with HIV than those who felt they had poor attitude (OR = 0.20; 95% CI: 0.07–0.57) (Table 3).

### Multivariable analysis of factors for delayed ART initiation

The adjusted analysis revealed that the odds of delayed ART initiation were significantly lower among respondents who were more than 35 years of age as compared to the odds of those who were less than or equal to 35 years of age (aOR = 0.35; 95% CI: 0.16–0.76). Similarly, the odds of delayed ART initiation among respondents who had no education were significantly lower than the odds of those who had attained a primary level of education (aOR = 0.04; 95% CI: 0.05–0.35). Also, the odds of delayed ART initiation among respondents who adhered to the

HIV clinic appointments were significantly lower as compared to the odds of their colleagues who did not adhere to the appointments (aOR = 0.06; 95% CI: 0.02–0.15). Further analysis showed a significantly lower odds of delayed ART initiation among those who lived in

**Table 1. Distribution of participants' characteristics, Alebtong district (N = 432).**

| Variables | Characteristics | Frequency (%) |
|---|---|---|
| Age of respondents | | |
| | ≤ 35 years | 293 (67.8) |
| | > 35 years | 139 (32.2) |
| Sex of respondent | | |
| | Male | 169 (39.1) |
| | Female | 263 (60.9) |
| Place of residence | | |
| | Rural | 405 (93.8) |
| | Urban | 27 (6.2) |
| Marital status | Not married | 69 (16.0) |
| | Married | 317 (73.4) |
| | Widow/widower | 22 (5.1) |
| | Separated/divorced | 24 (5.5) |
| Highest education | No education | 75 (17.4) |
| | Primary | 279 (64.6) |
| | Secondary | 69 (16.0) |
| | Post-secondary | 9 (2.1) |
| | No education | 75 (17.4) |
| Occupation | Subsistence farmer | 385 (80.6) |
| | Others* | 47 (19.4) |
| Religious affiliation | Catholics | 225 (52.1) |
| | Protestants | 169 (39.1) |
| | Others** | 38 (8.8) |
| SES of households | Low SES | 331 (76.6) |
| | Medium SES | 70 (7.2) |
| | High SES | 31 (7.2) |
| Disclosure of HIV status | Yes | 298 (69.0) |
| | No | 134 (31.0) |
| Given ARVs same day | Yes | 279 (64.6) |
| | No | 153 (35.4) |
| Distance to nearest h/f | ≤ 5 kilometres | 246 (56.9) |
| | > 5 kilometres | 186 (43.1) |
| Alcohol consumption | Yes | 93 (21.5) |
| | No | 339 (78.5) |
| Tobacco smoking | Yes | 41 (9.5) |
| | No | 391 (90.5) |
| Domestic violence | Never | 343 (79.4) |
| | Rarely | 40 (9.3) |
| | Sometimes | 49 (11.3) |
| **Total** | | **432** |

N = total number of participants, ART = antiretroviral therapy, HIV = human immunodeficiency virus, SES = socio-economic status, others* = formal & informal employment, others** = Muslim, Pentecostal & Seventh Day Adventists, & h/f = health facility.

**Table 2. ART initiation status at health facilities offering ART services in Alebtong district.**

| Health facility | Early ART n (%) | Delayed ART n (%) | Total (n) |
|---|---|---|---|
| Alebtong HC IV | 103 (81.1) | 24 (18.9) | 127 |
| Apala HC III | 77 (83.7) | 15 (16.3) | 92 |
| Abako HC III | 23 (76.7) | 7 (23.3) | 30 |
| Amugu HC III | 34 (89.5) | 4 (10.5) | 38 |
| Omoro HC III | 39 (86.7) | 6 (13.3) | 45 |
| Aloi Mission HC III | 48 (75.0) | 16 (25.0) | 64 |
| Alanyi Mission HC III | 30 (83.3) | 6 (16.7) | 36 |
| **Total (N)** | **354 (81.9)** | **78 (18.1)** | **432** |

ART = antiretroviral therapy, n = frequency, N = total number and HC = health centre.

households within the low socioeconomic status than those from the highest socioeconomic status (aOR = 0.27; 95% CI: 0.07–0.98). Also, significantly lower odds of delayed ART initiation were observed among respondents who were linked to care the same day they tested for HIV as compared to their counterparts (aOR = 0.21; 95% CI: 0.08–0.55). The odds of delayed ART initiation among respondents who never experienced stigma were about five times higher than the odds of those who experienced some form of stigma (aOR = 4.63; 95% CI: 1.48–14.46). Lastly, the odds of delayed ART initiation among respondents who said their culture does not support ART were approximately 11 times higher than the odds of those whose culture support ART (aOR = 10.62; 95% CI: 3.04–37.08). The overall final model shows a good fit of data in the model using the Pearson Chi-Square test with p = 0.745 and was able to successfully explain 52.0% of the variance in the outcome. Furthermore, it shows only 15.7 deviance after models were fitted indicating a great improvement in the final model as compared to models 1, 2, and 3 as shown in Table 4.

## Discussions

Our study examined the prevalence and factors associated with delayed ART initiation among adults living with HIV in Alebtong district, Northern Uganda. The study found the prevalence of delayed ART initiation among respondents to be 18.1%. Analysis of respondents' characteristics showed a higher proportion of respondents among female, rural dwellers, and younger respondents. At the multivariable analysis level, our key findings revealed a significantly lower odds of delayed ART initiation among respondents of more than 35 years, patients who adhered to HIV clinic appointments, and patients linked to care on the same day they tested for HIV. However, significantly higher odds of delayed ART initiation were observed among respondents whose cultures do not support the use of ART services.

This study revealed the prevalence of delayed ART initiation to be 18.1% which is much lower than the study undertaken in Western Ethiopia 34% [23]. Also, a systematic review and meta-analysis conducted in sub-Saharan Africa showed the prevalence of delayed ART initiation to be 78% in South Africa, 66% and 23% in Kenya respectively, and 67% in Ethiopia [24]. Potential explanations for the higher prevalence in other countries are; limited time of disclosure of HIV status that could potentially result in stigma, conflict, and domestic violence; uncertainty about the HIV test results, a desire for pregnant women to seek approval from their husbands before starting ART [25–28]. Other factors include a lack of belief in the healthcare benefits of early treatment and the use of substance abuse [24]. However, our study showed a low prevalence of delayed ART initiation due to good adherence to the recent test and treat policy, frequent follow-up of lost patients by the linkage facilitators who performs

**Table 3. Unadjusted analysis of factors and delayed ART initiation among adults with HIV.**

| Variable characteristics | ART Initiation | | Unadjusted OR (95% CI) |
|---|---|---|---|
| | Early ART n (%) | Delayed ART n (%) | |
| **Demographic factors** | | | |
| Age category (years) | | | |
| > 35 years (N = 139) | 107 (77.0) | 32 (23.0) | 0.61 (0.36–0.98) |
| ≤ 35 years (N = 293) | 247 (84.3) | 46 (15.7) | 1 |
| Sex of respondent | | | |
| Male (N = 169) | 137 (81.1) | 32 (18.9) | 1.10 (0.67–1.82) |
| Female (N = 263) | 217 (82.5) | 46 (17.5) | 1 |
| Place of residence | | | |
| Urban (N = 27) | 18 (66.7) | 9 (33.3) | 2.44* (1.05–5.65) |
| Rural (N = 405) | 336 (83.0) | 69 (17.0) | 1 |
| Current marital status | | | |
| Not married (N = 69) | 61 (88.4) | 8 (11.6) | 0.60 (0.27–1.32) |
| Widow/widower (N = 22) | 17 (77.3) | 5 (22.7) | 1.34 (0.48–3.79) |
| Separated/divorced (N = 24) | 16 (66.7) | 8 (33.3) | 2.28 (0.93–5.59) |
| Married (N = 317) | 260 (82.0) | 57 (18.0) | 1 |
| Highest education | | | |
| Post-secondary (N = 9) | 5 (55.6) | 4 (44.4) | 3.58 (0.93–13.79) |
| Secondary education (N = 69) | 62 (89.9) | 7 (10.1) | 0.51 (0.22–1.17) |
| No education (N = 75) | 59 (78.7) | 16 (21.3) | 1.21 (0.65–2.28) |
| Primary education (N = 279) | 228 (81.7) | 51 (18.3) | 1 |
| Occupation | | | |
| Subsistence farmer (N = 385) | 315 (81.8) | 70 (18.2) | 1.08 (0.49–2.42) |
| Others[a] (N = 47) | 39 (83.0) | 8 (17.0) | 1 |
| Religious affiliation | | | |
| Protestant (N = 169) | 135 (79.9) | 34 (20.1) | 1.32 (0.79–2.22) |
| Others[b] (N = 38) | 30 (78.9) | 8 (21.1) | 1.40 (0.59–3.30) |
| Catholic (N = 225) | 189 (84.0) | 36 (16.0) | 1 |
| **Individual factors** | | | |
| Time taken to h/f | | | |
| ≤ 60 minutes (N = 180) | 143 (79.4) | 37 (20.6) | 1.33 (0.81–2.18) |
| > 60 minutes (N = 252) | 211 (83.7) | 41 (16.3) | 1 |
| Frustrated with HIV result | | | |
| Yes (N = 240) | 187 (77.9) | 53 (22.1) | 1.89* (1.13–3.18) |
| No (N = 192) | 167 (87.0) | 25 (13.0) | 1 |
| Disclosure of HIV status | | | |
| Yes (N = 298) | 256 (85.9) | 42 (14.1) | 0.45** (0.27–0.74) |
| No (N = 134) | 98 (73.1) | 36 (26.9) | 1 |
| Adherence to h/f appointment | | | |
| Yes (N = 383) | 338 (88.3) | 45 (11.7) | 0.07*** (0.03–0.13) |
| No (N = 49) | 16 (32.7) | 33 (67.3) | 1 |
| Drink alcohol | | | |
| No (N = 339) | 290 (85.5) | 49 (14.5) | 0.37** (0.19–0.76) |
| Yes (N = 93) | 64 (68.8) | 29 (31.2) | 1 |
| Tobacco smoking | | | |
| No (N = 391) | 327 (83.6) | 64 (16.4) | 1.38** (1.32–5.33) |
| Yes (N = 41) | 27 (65.9) | 14 (34.1) | 1 |

(*Continued*)

**Table 3.** (*Continued*)

| Variable characteristics | ART Initiation | | Unadjusted OR (95% CI) |
|---|---|---|---|
| | **Early ART n (%)** | **Delayed ART n (%)** | |
| Satisfaction with ART | | | |
| Most/all the time (N = 153) | 133 (86.9) | 20 (13.1) | 0.62 (0.33–1.16) |
| Sometimes (N = 135) | 105 (77.8) | 30 (22.2) | 1.18 (0.66–2.11) |
| Never/rarely (N = 144) | 116 (80.6) | 28 (19.4) | 1 |
| **Community factors** | | | |
| Distance to nearest h/f | | | |
| > 5 Kilometres (N = 186) | 157 (84.4) | 29 (15.6) | 0.74 (0.45–1.23) |
| ≤ 5 Kilometres (N = 246) | 197 (80.1) | 49 (19.9) | 1 |
| SES of households | | | |
| Low SES (N = 331) | 283 (85.5) | 48 (14.5) | 0.27*** (0.12–0.59) |
| Medium SES (N = 70) | 52 (74.3) | 18 (25.7) | 0.55 (0.22–1.35) |
| High SES (N = 31) | 19 (61.3) | 12 (38.7) | 1 |
| Domestic violence | | | |
| Never (N = 343) | 287 (83.7) | 56 (16.3) | 0.49* (0.25–0.97) |
| Rarely (N = 40) | 32 (80.0) | 8 (20.0) | 0.63 (0.23–1.69) |
| Sometimes (N = 49) | 35 (71.4) | 14 (28.6) | 1 |
| Experienced stigma | | | |
| Never (N = 88) | 77 (87.5) | 11 (12.5) | 1.60 (0.79–3.23) |
| Rarely (N = 264) | 215 (81.4) | 49 (18.6) | 2.03 (0.89–4.62) |
| Sometimes (N = 80) | 62 (77.5) | 18 (22.5) | 1 |
| Cultural support on ART | | | |
| No (N = 21) | 8 (38.1) | 13 (61.9) | 8.02*** (3.18–20.19) |
| Don't know (N = 61) | 55 (90.2) | 6 (9.8) | 0.54 (0.22–1.31) |
| Yes (N = 350) | 291 (83.1) | 59 (16.9) | 1 |
| Health services factors | | | |
| HIV test from same h/f | | | |
| Yes (N = 369) | 313 (84.8) | 56 (15.2) | 0.33*** (0.19–0.60) |
| No (N = 63) | 41 (65.1) | 22 (34.9) | 1 |
| Linked to care from | | | |
| Outreach services (N = 34) | 15 (44.1) | 19 (55.9) | 7.10*** (3.40–14.84) |
| Maternity ward (N = 26) | 23 (88.5) | 3 (11.5) | 0.73 (0.21–2.52) |
| Antenatal clinic (N = 22) | 19 (86.4) | 3 (13.6) | 0.89 (0.25–3.10) |
| Outpatient department (N = 350) | 297 (84.9) | 53 (15.1) | 1 |
| Given HIV result same day tested | | | |
| Yes (N = 421) | 347 (82.4) | 74 (17.6) | 0.37 (0.11–1.31) |
| No (N = 11) | 7 (63.6) | 4 (36.4) | 1 |
| Linkage to care same day tested | | | |
| Yes (N = 359) | 313 (87.2) | 46 (12.8) | 0.19*** (0.11–0.33) |
| No (N = 73) | 41 (56.2) | 32 (43.8) | 1 |
| Attitude of health workers | | | |
| Very good (N = 106) | 91 (85.8) | 15 (14.2) | 0.20** (0.07–0.57) |
| Good (N = 208) | 172 (82.7) | 36 (17.3) | 0.26** (0.10–0.66) |
| Fairly good (N = 98) | 80 (81.6) | 18 (18.4) | 0.28** (0.10–0.76) |

(*Continued*)

**Table 3.** (Continued)

| Variable characteristics | ART Initiation | | Unadjusted OR (95% CI) |
|---|---|---|---|
| | Early ART n (%) | Delayed ART n (%) | |
| Poor (N = 20) | 11 (55.0) | 9 (45.0) | 1 |

OR = odds ratio

***P<0.001

**P<0.01

*P<0.05

CI = confidence interval, n = frequency, HIV = human immune-deficiency virus, h/f = health facility, no. = number, ART = antiretroviral therapy, others[a] = informal & formal employment, others[b] = Muslim, Born Again, Orthodox & Seventh Day Adventist and SE = socio-economic status.

counselling and referral of people living with HIV from the community level to the health facility level.

Our study revealed that respondents who were more than 35 years had significantly lower odds of delayed ART initiation. In contrast, a study conducted in Ethiopia revealed that adults with HIV who were between the age group of 35–44 years had significantly higher odds of delayed ART initiation [29]. Similarly, a study conducted in Kenya observed that older patients had higher odds of delayed ART initiation [30]. The potential explanations for the differences are that older persons are at lower risk of acquiring HIV because they are more informed about HIV issues due to the fact that they are exposed to or able to acquire phones, radios, television, newspapers, and other social media sources where HIV information can be found. Additionally, they have good healthcare seeking behavior so they frequently interact with healthcare workers during consultations, antenatal care visits, delivery, vaccination, and during postnatal care services [30]. On the other hand, young patients are more prone to devastating effects of HIV due to peer pressure, community stigmatization, and fear of criticism from the society and health care workers leading to poor healthcare seeking behaviour, limited access to information since they are unable to acquire phones, radio, television, and other social media that could be the source of HIV information. This means that more emphasis on HIV services should be geared towards adolescents and young people.

This study further showed that respondents who adhered to HIV clinic appointments had significantly lower odds of delayed ART initiation. The benefits of adherence to clinic attendance are; routine laboratory investigations, early clinician assessments, and prompt treatment, and provides an opportunity for primary caregivers to rapidly detect any co-morbidity. However, most studies conducted in Africa showed a lack of adherence to clinic appointments as a reason for delayed ART initiation [2–34]. The possible explanation for differences could be that our healthcare workers are committed and always available on duty, have good patient care, provide good information to patients and patients believe in the healthcare systems. So the motivated patients would try as much as possible to adhere to their clinic appointments. Reasons for non-adherence to clinic appointments are; transfer to another clinic [35], and financial reasons including the high cost of transportation and clinic visits [36]. Another study in Kenya noted that women cited family commitments for missing clinic appointments and men cited work commitments [37]. Besides, the non-availability of some drugs and the perceived rudeness of some clinic staff frustrated and discouraged persons living with HIV from seeking ART care services [38].

Our study found a lack of cultural support as being a significant predictor of delayed ART initiation. Cultural support in this context refers to social thoughts, beliefs, manners, and actions that influence the use of ART. The social beliefs and actions can lead to early or delayed

**Table 4. Adjusted analysis of factors and delayed ART initiation among adults with HIV.**

| Variable Characteristics | Delayed ART n (%) | Model 1 aOR (95%CI) | Model 2 aOR (95%CI) | Model 3 aOR (95%CI) | Model 4 aOR (95%CI) |
|---|---|---|---|---|---|
| **Demographic factors** | | | | | |
| Age category (years) | | | | | |
| > 35 years (N = 139) | 32 (23.0) | 0.64 (0.37–1.11) | 0.53 (0.28–1.01) | 0.40*(0.19–0.82) | 0.35*(0.16–0.76) |
| ≤ 35 years (N = 293) | 46 (15.7) | 1 | 1 | 1 | 1 |
| Sex of respondent | | | | | |
| Male (N = 169) | 32 (18.9) | 0.81 (0.47–1.41) | 1.12 (0.57–2.23) | 1.40 (0.65–3.03) | 1.69 (0.74–3.85) |
| Female (N = 263) | 46 (17.5) | 1 | 1 | 1 | 1 |
| Place of residence | | | | | |
| Urban (N = 27) | 9 (33.3) | 0.37* (0.15–0.90) | 0.78 (0.24–2.49) | 0.71 (0.20–2.54) | 0.33 (0.08–1.31) |
| Rural (N = 405) | 69 (17.0) | 1 | 1 | 1 | 1 |
| Marital status | | | | | |
| Not married (N = 69) | 8 (11.6) | 0.62 (0.27–1.41) | 0.50 (0.20–1.29) | 0.40 (0.13–1.23) | 0.40 (0.12–1.37) |
| Widow/widower (N = 22) | 5 (22.7) | 1.02 (0.34–3.08) | 0.62 (0.17–2.18) | 0.66 (0.16–2.77) | 0.60 (0.13–2.77) |
| Separated/divorced (N = 24) | 8 (33.3) | 2.56 (0.99–6.61) | 1.96 (0.62–6.19) | 2.24 (0.68–7.42) | 2.09 (0.62–7.06) |
| Married (N = 317) | 57 (18.0) | 1 | 1 | 1 | 1 |
| Highest education | | | | | |
| Post-secondary (N = 9) | 4 (44.4) | 0.33 (0.07–1.52) | 0.16*(0.03–0.84) | 0.16*(0.03–0.96) | 0.19 (0.02–1.57) |
| Secondary education (N = 69) | 7 (10.1) | 0.27 (0.06–1.13) | 0.14**(0.03–0.64) | 0.14*(0.03–0.73) | 0.17 (0.02–1.23) |
| No education (N = 75) | 16 (21.3) | 0.11**(0.02–0.54) | 0.04**(0.01–0.24) | 0.03**(0.01–0.22) | 0.04**(0.02–0.35) |
| Primary education (N = 279) | 51 (18.3) | 1 | 1 | 1 | 1 |
| Occupation | | | | | |
| Subsistence farmer (N = 385) | 70 (18.2) | 0.93 (0.36–2.41) | 2.20 (0.65–7.50) | 3.80 (0.92–15.74) | 2.92 (0.61–14.04) |
| Others[a] (N = 47) | 8 (17.0) | 1 | 1 | 1 | 1 |
| Religion | | | | | |
| Protestant (N = 169) | 34 (20.1) | 0.63 (0.26–1.51) | 0.50 (0.18–1.36) | 0.48 (0.15–1.51) | 0.52 (0.16–1.68) |
| Others[b] (N = 38) | 8 (21.0) | 0.73 (0.30–1.79) | 0.51 (0.19–1.42) | 0.46 (0.14–1.52) | 0.53 (0.16–1.80) |
| Catholic (N = 225) | 36 (16.0) | 1 | 1 | 1 | 1 |
| **Individual factors** | | | | | |
| Time taken to h/f | | | | | |
| ≤ 60 minutes (N = 180) | 37 (20.6) | | 0.56 (0.29–1.06) | 0.82 (0.39–1.82) | 1.10 (0.48–2.51) |
| > 60 minutes (N = 252) | 41 (16.3) | | 1 | 1 | 1 |
| Frustration | | | | | |
| Yes (N = 240) | 53 (22.1) | | 1.84 (0.97–3.49) | 2.18* (1.04–4.59) | 2.01 (0.94–4.30) |
| No (N = 192) | 25 (13.0) | | 1 | 1 | 1 |
| Disclosure of status | | | | | |
| Yes (N = 298) | 42 (14.1) | | 0.58 (0.31–1.11) | 0.62 (0.30–1.30) | 0.66 (0.30–1.47) |
| No (N = 134) | 36 (26.9) | | 1 | 1 | 1 |
| Adherence to appt | | | | | |
| Yes (N = 383) | 45 (11.7) | | 0.07**(0.03–0.15) | 0.04**(0.01–0.09) | 0.06***(0.02–0.15) |
| No (N = 49) | 33 (67.3) | | 1 | 1 | 1 |
| Drink alcohol | | | | | |
| No (N = 339) | 49 (14.5) | | 0.49 (0.19–1.31) | 0.91 (0.28–3.00) | 1.04 (0.29–3.69) |
| Yes (N = 93) | 29 (31.2) | | 1 | 1 | 1 |
| ART Satisfaction | | | | | |
| Most/all the time (N = 153) | 20 (13.1) | | 0.86 (0.38–1.92) | 0.71 (0.29–1.73) | 0.79 (0.29–2.17) |
| Sometimes (N = 135) | 30 (22.2) | | 0.92 (0.44–1.94) | 1.18 (0.50–2.81) | 1.24 (0.50–3.10) |
| Never/rarely (N = 144) | 28 (19.4) | | 1 | 1 | 1 |
| **Community factors** | | | | | |

(*Continued*)

**Table 4.** (Continued)

| Variable Characteristics | Delayed ART n (%) | Model 1 aOR (95%CI) | Model 2 aOR (95%CI) | Model 3 aOR (95%CI) | Model 4 aOR (95%CI) |
|---|---|---|---|---|---|
| Distance to h/f | | | | | |
| > 5 Kilometres (N = 186) | 29 (15.6) | | | 0.65 (0.29–1.46) | 1.03 (0.41–2.60) |
| ≤ 5 Kilometres (N = 246) | 49 (19.9) | | | 1 | 1 |
| SES of households | | | | | |
| Low SES (N = 331) | 48 (14.5) | | | 0.16** (0.05–0.51) | 0.27* (0.07–0.98) |
| Medium SES (N = 70) | 18 (25.7) | | | 0.41 (0.12–1.46) | 0.60 (0.16–2.31) |
| High SES (N = 31) | 12 (38.7) | | | 1 | 1 |
| Domestic violence | | | | | |
| Never (N = 343) | 56 (16.3) | | | 0.48 (0.18–1.27) | 0.48 (0.16–1.40) |
| Rarely (N = 40) | 8 (20.0) | | | 0.45 (0.12–1.73) | 0.37 (0.08–1.62) |
| Sometimes (N = 49) | 14 (28.6) | | | 1 | 1 |
| Experienced stigma | | | | | |
| Never (N = 88) | 11 (12.5) | | | 4.00** (1.39–11.53) | 4.63** (1.48–14.26) |
| Rarely (N = 264) | 49 (18.6) | | | 2.99 (0.95–9.43) | 3.47* (1.02–11.79) |
| Sometimes (N = 80) | 18 (22.5) | | | 1 | 1 |
| Cultural support on ART | | | | | |
| No (N = 21) | 13 (61.9) | | | 10.84* (3.16–13.19) | 10.62** (3.04–14.08) |
| Don't know (N = 61) | 6 (9.8) | | | 0.35 (0.11–1.18) | 0.28 (0.07–1.09) |
| Yes (N = 350) | 59 (16.9) | | | 1 | 1 |
| **Health factors** | | | | | |
| Test from same h/f | | | | | |
| Yes (N = 369) | 56 (15.2) | | | | 1.85 (0.55–6.23) |
| No (N = 63) | 22 (34.9) | | | | 1 |
| Linked to care from | | | | | |
| Outreach services (N = 34) | 19 (55.9) | | | | 2.54 (0.71–9.15) |
| Maternity ward (N = 26) | 3 (11.5) | | | | 0.54 (0.10–2.90) |
| Antenatal clinic (N = 22) | 3 (13.6) | | | | 1.04 (0.18–5.94) |
| Outpatient dept (N = 350) | 53 (15.1) | | | | 1 |
| Given HIV result same day | | | | | |
| Yes (N = 421) | 74 (17.6) | | | | 0.72 (0.12–4.12) |
| No (N = 11) | 4 (36.4) | | | | 1 |
| Linked to care the same day | | | | | |
| Yes (N = 359) | 46 (12.8) | | | | 0.21** (0.08–0.55) |
| No (N = 73) | 32 (43.8) | | | | 1 |
| Attitude of h/workers | | | | | |
| Very good (N = 106) | 15 (14.2) | | | | 0.36 (0.07–1.81) |
| Good (N = 208) | 36 (17.3) | | | | 0.26 (0.06–1.16) |
| Fairly good (N = 98) | 18 (18.4) | | | | 0.30 (0.07–1.35) |
| Poor (N = 20) | 9 (45.0) | | | | 1 |
| Model fitness test, $x^2$ | | p = 0.886 | p = 0.246 | p = 0.467 | p = 0.745 |
| Nagelkereke pseudo, $r^2$ | | 8.2% | 34.0% | 47.8% | 52.0% |
| Log-Likelihood, $x^2$ | | 22.3 | 78.4 | 48.6 | 15.7 |

OR = odds ratio

***P<0.001

**P<0.01

*P<0.05

CI = confidence interval, $x^2$ = Pearson chi-square, n = frequency, HIV = human immune-deficiency virus, h/f = health facility, ART = antiretroviral therapy, others[a] = informal & formal employment, others[b] = Muslim, Born Again, Orthodox, Seventh Day Adventist and SE = socio-economic status, appt = appointment, and dept = department.

ART initiation by persons living with HIV in a particular society. A study in Ethiopia found a four-fold increase in delayed ART initiation among persons with HIV who used traditional medicines as an alternative to ART [39]. Similarly, a study conducted in Uganda reported that persons living with HIV who had received care from non-medical providers were more likely to delay ART initiation [40]. Some authors have suggested that people are more likely to engage in care and benefit from the most optimal health behaviours if their cultural beliefs are accepted in the health care system. Although research is limited in this area, a study among persons with HIV in Puerto Ricans in the US found higher ART adherence levels among those acculturated in the US culture [41]. Many factors may contribute to the patient's decision to use traditional treatments. Some of them could be strong cultural beliefs, perception of traditional healers being easily accessible, and experience of lack of cure from using ART [42]. In our community, there are mainly three kinds of traditional healers who claim they can cure persons with HIV-related illnesses. These are witchdoctors, herbalists, and spiritualists. Witchdoctors use water, ash, salt, candles, oil, animals/birds, and other materials to intervene and communicate with the spiritual world on behalf of their patients. Herbalists use indigenous plants, water, bones, and stones to provide remedies for individuals with whom they have diagnosed themselves or those that are referred for specialized care by other herbalists, friends, and relatives. Spiritualists use Godly spirits and ancestors to learn about people's problems. People who use traditional medicine have a strong conviction that illnesses can be cured by only visiting traditional healers. More interventions need to be identified to address negative cultural beliefs and norms related to ART services.

Lastly, our study found that patients linked to care on the same day they are tested for HIV had significantly lower odds of delayed ART initiation. Our finding is similar to a study conducted in Zambia and South Africa which found that the time from linkage to care to ART initiation was short, with a median of less than 1 month, while the time from referral to linkage to care was longer, with a median of 4 months [43]. A potential explanation for the similarity could be the benefits associated with linkage to care same day tested. Some of the enablers are; a desire to prevent further transmission of HIV, supportive counselling, and a perception that starting early ART would possibly reduce the risk of stigma [25–27, 44]. Further explanations could be that health facilities have better-trained health staff, better physical infrastructure that offers privacy, confidentiality, and a better level of comfort to treat persons with HIV [45].

## Strengths and limitations

Some of the strengths registered in our study were: the high response rate of 96.9% and the use of a variety of data sources to recruit participants which could have widely reduced biases in the study. This is the first study of its kind in the northern region of Uganda; therefore, the findings can be generalized to other districts in the region due to similarities in the demographic profiles and healthcare systems delivery. Furthermore, the findings of our study can be translated into HIV treatment and care policy in Uganda. However, the study had some limitations which include self-reporting by respondents which could have introduced information bias and the possibility of selection bias as a result of having the majority of respondents in the young age group. Also, there is a possibility of some variables we controlled for confounders were mediation factors. However, our study did not take care of mediation factors. The findings of our study may apply to other resource-limited settings in Northern Uganda.

## Conclusions

Our study was conducted in Alebtong district for a period of three months (March-May 2018), it aimed to examine the prevalence and factors associated with delayed ART initiation

among persons living with HIV in the previous 12 months, April 2017 to March 2018. The results found the overall prevalence of delayed ART initiation to be 18.1%. After final adjustment, our key findings revealed a significantly lower odds of delayed ART initiation among respondents of more than 35 years, patients who adhered to HIV clinic appointments, and patients linked to care on the same day tested for HIV. A significantly higher odds of delayed ART initiation were observed among respondents whose cultures do not support the use of ART services. Improving early ART initiation in the district requires strengthening the involvement of adolescents and young people in the HIVAIDS programming and identifying strategies to enhance adherence to HIV clinic appointments. The district needs to further scale up the implementation of the same-day ART initiation policy, and evaluate and address negative cultural beliefs and norms affecting early ART initiation in the community.

## Supporting information

**S1 Text. Structured questionnaire.**
(PDF)

**S2 Text. Original structured questionnaire.**
(PDF)

**S1 Table. Data extraction tool.**
(PDF)

**S1 Fig. Flow chart showing sampling procedure.**
(PDF)

## Acknowledgments

We appreciate all the research supervisors, research assistants, and study participants. The district health officer of Alebtong district, and Makerere University School of Public Health, Kampala, Uganda for approving this study.

## Author Contributions

**Conceptualization:** Anthony Mark Ochen, Victoria Nankabirwa.

**Data curation:** Anthony Mark Ochen, Michael Ediau, Victoria Nankabirwa.

**Formal analysis:** Anthony Mark Ochen, David Lubogo, Michael Ediau, Victoria Nankabirwa.

**Investigation:** Anthony Mark Ochen, David Lubogo.

**Methodology:** Anthony Mark Ochen, David Lubogo, Michael Ediau.

**Project administration:** Anthony Mark Ochen, David Lubogo, Victoria Nankabirwa.

**Resources:** David Lubogo, Victoria Nankabirwa.

**Software:** Anthony Mark Ochen.

**Supervision:** Michael Ediau, Victoria Nankabirwa.

**Validation:** David Lubogo, Michael Ediau, Victoria Nankabirwa.

**Writing – original draft:** Anthony Mark Ochen.

**Writing – review & editing:** David Lubogo, Michael Ediau, Victoria Nankabirwa.

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
