## [Decision Letter · Decision Letter 0]

25 Jan 2022

PGPH-D-21-01005

Prevalence and Factors Associated with Delayed Antiretroviral Therapy Initiation among HIV positive adults in Alebtong District, Northern Uganda: a facility-based study

Dear Dr. Ochen,

Thank you for submitting your manuscript to PLOS Global Public Health. After careful consideration, we feel that it has merit but does not fully meet PLOS Global Public Health’s publication criteria as it currently stands. Therefore, we invite you to submit a revised version of the manuscript that addresses the points raised during the review process. Every reviewer comment is appropriate and addressing all of them may allow us to consider your paper for publication in our journal. Most comments are focused on improving the presentation of results and improving the discussion, but there are also comments regarding grammar, style, and overall presentation. 

We look forward to receiving your revised manuscript.

Kind regards,

Kevin Escandón, MD, MSc

Academic Editor

Journal Requirements:

1. Please include additional information regarding the survey or questionnaire used in the study and ensure that you have provided sufficient details that others could replicate the analyses. For instance, if you developed a questionnaire as part of this study and it is not under a copyright more restrictive than CC-BY, please include a copy, in both the original language and English, as Supporting Information.

2. Please amend your Financial Disclosure statement. If you did not receive any funding for this study, please simply state: “The authors received no specific funding for this work.”

3. Please update your Competing Interests statement. If you have no competing interests to declare, please state: “The authors have declared that no competing interests exist.”

4. Please provide a complete Data Availability Statement in the submission form. If your research concerns only data provided within your submission, please write “All data are in the manuscript and/or supporting information files.” as your Data Availability Statement.

5. We have noticed that you have a list of supporting information legends in your manuscript. However, there are no corresponding files uploaded to the submission. Please upload them as separate files with the file type 'Supporting Information'. Please ensure that all Supporting Information files are included correctly and that each one has a legend listed in the manuscript after the references list.

Reviewers' comments:

Reviewer's Responses to Questions

**Comments to the Author**

1. Does this manuscript meet PLOS Global Public Health’s publication criteria? Is the manuscript technically sound, and do the data support the conclusions? The manuscript must describe methodologically and ethically rigorous research with conclusions that are appropriately drawn based on the data presented.

Reviewer #1: Yes

Reviewer #2: Partly

2. Has the statistical analysis been performed appropriately and rigorously?

Reviewer #1: No

Reviewer #2: No

3. Have the authors made all data underlying the findings in their manuscript fully available (please refer to the Data Availability Statement at the start of the manuscript PDF file)?

Reviewer #1: No

Reviewer #2: Yes

4. Is the manuscript presented in an intelligible fashion and written in standard English?

Reviewer #1: Yes

Reviewer #2: No

5. Review Comments to the Author

Reviewer #1: This was an interesting study detailing the factors associated with delayed ART initiation among people living with HIV in northern Uganda. I have identified several areas of the manuscript that the authors could strengthen.

The authors should be using people first language. HIV-infected/HIV-positive/HIV patients are no longer acceptable terms to use in the literature due to their stigmatizing effects. Please use ‘person living with HIV’ or ‘person with HIV.’

Abstract: inaccurate spell out of HIV abbreviation and no spell out of AIDS abbreviation in introduction. Should use updated stats if possible. The conclusion should not simply repeat the results but rather it should consider the future implications of the findings in the context of prior literature.

The data availability statement needs revising. The data are not in the text. Did the authors mean to say that the instructions for accessing the data are included in the text?

Introduction – need to define delayed ART initiation sooner; what is the cost of ART in Uganda?

Materials and Methods

Study context: Prevalence estimates are inconsistent with the introduction

Line 252 do you mean ‘registries’ rather than ‘registers’?

Lines 282-288: excessive and unimportant info included about the choice of method. Recommend shortening.

Line 295: states models 1, 3 and 4 have 7 demographic vars, but model 2 only has 6. What demographic variable was removed from model 2 and why?

The rationale for trying each of these models is needed. Why is block 1 always included but block 4 variables are only tested in the final model? What about the other possible combinations? Including those that break apart the “blocks” into smaller components? Did you consider a variable selection procedure that maximizes model performance while penalizing for the number of variables included? (e.g. AIC or BIC)

I appreciate the VIF approach used to reducing issues of collinearity. Did you consider whether some variables might be colliders in each model? It would be helpful to have a DAG since there are a lot of variables being tested.

Line 299 states that only one model was discussed. Why?

Did you perform p-value correction for multiple testing in the unadjusted analysis?

Results

The authors use ‘odds’ and ‘likelihoods’ interchangeably when they should not be.

Line 324: a higher proportion of what?

There appear to be errors in Table 3. Some of the ORs are not within the confidence intervals [e.g., age 1.61 (0.36-0.98), tobacco smoking 0.38** (1.32-5.33)].

Setting the reference group for the stigma variable to “never” would increase the interpretability of this variable. Just a thought.

Discussion:

The prevalence of delayed ART initiation is lower in this cohort than is expected from this region according to the studies cited in the introduction. What might be the reasons for this? Is there something about this particular sample/cohort (e.g., better care, better follow-up, etc.) that makes this study population different from the target population?

Lines 416-417: ‘….adult HIV patients who were in the age group between 35-44 years were 3.85 times higher to delay ART initiation…’ this statement is confusing. Do you mean the odds of delayed initiation were higher in this age group?

Why do you suppose the odds of delayed ART initiation were higher among urban vs rural residents in the unadjusted analysis but not in the adjusted analysis?

There were a few other findings that need more explanation:

- Why was ‘no education’ protective compared to ‘primary education’?

- Why was stigma protective?

Were these variables truly protective or could the model have been misspecified (e.g., some variables controlled for were actually colliders or other variables were not available)?

The emphasis on cultural acceptance, one of the major findings of this study, is essential. The authors include a detailed discussion on this topic, but the readers are left wondering what can be done about this. Are there any successful approaches available in the literature that can be cited?

Reviewer #2: Thank you for the opportunity to review this manuscript. This study adds relevant data to the literature on delayed ART initiation, which remains timely. The analysis appears to have largely been performed appropriately, though I have some concerns about the inclusion criteria as well as the multivariable models presented, noted below. The writing needs substantial revision, both in terms of framing the study in the Introduction and Discussion, and some general copy editing.

MAJOR COMMENTS

The introduction is too long and could be better organized - currently, there are many statements presented that seem quite peripheral to the matter of delayed ART initiation, the focus jumps from Uganda to sub-Saharan Africa to the global pandemic, and the authors do not provide a clear narrative about why their study is important and necessary. I would encourage the authors to reframe the introduction to clearly describe the scope of the problem, what is known about it, and what is unknown (i.e. the gap that their study is addressing) and only retain the necessary parts of the text to communicate this.

The sections in the introduction that describe the HIV pandemic and delivery of HIV care in Uganda (much of the text from lines 138-184) can be moved to Methods (Study Context subsection) - would also try to shorten this text substantially.

It is not entirely clear to me who was included in the study population, described in their sampling strategy in the Study Design and Sampling Procedure section. The authors present an N of 831 as the “population size” - unclear what, was this all newly diagnosed PLWH receiving care in the study centers? They also note on lines 213-215 that “all individuals tested…were included in the study.” Can they provide a clearer description of the OVERALL sample from which the study population was derived?

Did patients need to have started ART in order to be included in the study? From the text in lines 270-273 it seems that patients who never started ART were included in the “Delayed ART initiation” group. I would recommend that patients that never initiated ART be their own group or not included in the study.

The authors could provide a little more clarity with respect to some of the covariates included in the study: 1) what is “frustration with HIV results?;” 2) why include both time to reach health facility (as an individual characteristic) and distance to health facility (as a community characteristic)? - these seem redundant; 3) why is experience of stigma a community characteristic rather than an individual one?

The authors provide criteria for inclusion of variables in the multivariable analysis in the methods (288-290), but it seems as if all of the variables were included in both bivariate and multivariate models, even though some were clearly not significant in bivariate analyses and seem peripherally related to ART initiation. Can the authors either justify this or present a more streamlined multivariate analysis?

Are the results presented in Table 4 accurate? For age >35 and no education, higher proportions of patients had delayed ART starts, but the odds of delayed ART were significantly lower. Similarly, for patients that never experienced stigma, a lower proportion of patients had delayed ART start, but odds of delayed ART were higher. This is problematic.

The Discussion is too long, and the points that the authors are making are not always well-contextualized. Some of the literature that they choose for comparison is old and the comparisons are thus not entirely valid. The interpretations about why their results differ from others sometimes feels speculative (for example, lines 448-451). I would recommend that the authors first check their models to ensure accuracy (see prior point), the choose 3 of the most important findings, and then carefully review the literature not only for prior findings but for potential reasons why there may be differences or similarities.

The authors note that the observed rate of delayed ART is much lower than previous studies, however, they fail to note that 3 of the 4 studies they cite were from the pre-Treat All era. Currently and recently (i.e. during the study period), it is commonplace for patients to initiate ART on the day of diagnosis or very shortly thereafter. Therefore, they should make comparisons to studies conducted within the last 4-5 years, after global Treat All implementation.

MINOR COMMENTS

No need to spell out HIV or AIDS, these are universally understood acronyms.

What is HMIS (line 213)?

Lines 282-288 describing logistic regression are not necessary.

Instead of writing “less likely odds of…” I would suggest “lower odds of….”

Lines 411-414 in the Discussion are very unclear.

Overall, the manuscript requires careful proofreading for grammar, spelling, punctuation and clarity.

6. PLOS authors have the option to publish the peer review history of their article (what does this mean?). If published, this will include your full peer review and any attached files.

**Do you want your identity to be public for this peer review?** For information about this choice, including consent withdrawal, please see our Privacy Policy.

Reviewer #1: No

Reviewer #2: **Yes: **Jonathan Ross

---

## [Decision Letter · Decision Letter 1]

14 Jun 2022

PGPH-D-21-01005R1

Prevalence and factors associated with delayed Antiretroviral Therapy initiation among adults with HIV in Alebtong district, Northern Uganda: a facility-based study

Dear Dr. Ochen,

Thank you for submitting your manuscript to PLOS Global Public Health. After careful consideration, we feel that it has merit but does not fully meet PLOS Global Public Health’s publication criteria as it currently stands. Therefore, we invite you to submit a revised version of the manuscript that addresses the points raised during the review process.

We look forward to receiving your revised manuscript.

Kind regards,

Lei Gao

Academic Editor

Journal Requirements:

Additional Editor Comments (if provided):

Reviewers' comments:

Reviewer's Responses to Questions

**Comments to the Author**

1. If the authors have adequately addressed your comments raised in a previous round of review and you feel that this manuscript is now acceptable for publication, you may indicate that here to bypass the “Comments to the Author” section, enter your conflict of interest statement in the “Confidential to Editor” section, and submit your "Accept" recommendation.

Reviewer #2: (No Response)

2. Does this manuscript meet PLOS Global Public Health’s publication criteria? Is the manuscript technically sound, and do the data support the conclusions? The manuscript must describe methodologically and ethically rigorous research with conclusions that are appropriately drawn based on the data presented.

Reviewer #2: Yes

3. Has the statistical analysis been performed appropriately and rigorously?

Reviewer #2: Yes

4. Have the authors made all data underlying the findings in their manuscript fully available (please refer to the Data Availability Statement at the start of the manuscript PDF file)?

Reviewer #2: Yes

5. Is the manuscript presented in an intelligible fashion and written in standard English?

Reviewer #2: Yes

6. Review Comments to the Author

Reviewer #2: See attached for comments

7. PLOS authors have the option to publish the peer review history of their article (what does this mean?). If published, this will include your full peer review and any attached files.

**Do you want your identity to be public for this peer review?** For information about this choice, including consent withdrawal, please see our Privacy Policy.

Reviewer #2: **Yes: **Jonathan Ross

---

## [Editor Report · Decision Letter 2]

18 Jul 2022

Prevalence and factors associated with delayed Antiretroviral Therapy initiation among adults with HIV in Alebtong district, Northern Uganda: a facility-based study

PGPH-D-21-01005R2

Dear Mr. Ochen,

We are pleased to inform you that your manuscript 'Prevalence and factors associated with delayed Antiretroviral Therapy initiation among adults with HIV in Alebtong district, Northern Uganda: a facility-based study' has been provisionally accepted for publication in PLOS Global Public Health.

Best regards,

Lei Gao

Academic Editor